# Germline variant burden in cancer genes correlates with age at diagnosis and somatic mutation burden

Tao Qing[1], Hussein Mohsen [2], Michal Marczyk [1,3], Yixuan Ye [2], Tess O'Meara[1], Hongyu Zhao [2,4], Jeffrey P. Townsend [2,4], Mark Gerstein[2,5,6,7], Christos Hatzis[1,10], Yuval Kluger [2,8,9] & Lajos Pusztai [1✉]

Cancers harbor many somatic mutations and germline variants, we hypothesized that the combined effect of germline variants that alter the structure, expression, or function of protein-coding regions of cancer-biology related genes (gHFI) determines which and how many somatic mutations (sM) must occur for malignant transformation. We show that gHFI and sM affect overlapping genes and the average number of gHFI in cancer hallmark genes is higher in patients who develop cancer at a younger age ($r = -0.77$, $P = 0.0051$), while the average number of sM increases in increasing age groups ($r = 0.92$, $P = 0.000073$). A strong negative correlation exists between average gHFI and average sM burden in increasing age groups ($r = -0.70$, $P = 0.017$). In early-onset cancers, the larger gHFI burden in cancer genes suggests a greater contribution of germline alterations to the transformation process while late-onset cancers are more driven by somatic mutations.

[1] Breast Medical Oncology, School of Medicine, Yale University, New Haven, CT, USA. [2] Computational Biology and Bioinformatics Program, Yale University, New Haven, CT, USA. [3] Data Mining Division, Silesian University of Technology, Gliwice, Poland. [4] Department of Biostatistics, School of Public Health, Yale University, New Haven, CT, USA. [5] Department of Molecular Biophysics & Biochemistry, Yale University, New Haven, CT, USA. [6] Department of Computer Science, Yale University, New Haven, CT, USA. [7] Department of Statistics & Data Science, Yale University, New Haven, CT, USA. [8] Department of Pathology, School of Medicine, Yale University, New Haven, CT, USA. [9] Program of Applied Mathematics, Yale University, New Haven, CT, USA. [10]Present address: Bristol-Myers Squibb, New York, NY, USA. ✉email: lajos.pusztai@yale.edu

Approximately 40% of the population living in the USA will develop cancer during their lifetime[1,2]. The transformation of a normal cell to a cancer cell requires many genomic and epigenetic alterations in key cellular metabolic and regulatory processes that are collectively referred to as the hallmarks of cancer[3]. Family history and increasing age are the two most consistent risk factors for cancer, suggesting that both inherited and acquired events contribute to cancer development[4,5]. The higher incidence of cancer in older individuals is attributed to accumulation of acquired somatic mutations (sMs) and epigenetic changes during lifetime, that can be accelerated by environmental factors in various tissues[6]. Large-scale cancer genome sequencing projects have confirmed that most cancers harbor a few highly recurrent sMs in classical oncogenes and tumor suppressor genes (e.g. *p53*, *PIK3CA*) along with many more individually rare sMs in unique combinations[7]. All individuals also carry hundreds of common and rare (at population level) high-functional impact germline variants that contribute to our susceptibilities for disease and account for heritability of cancer[8]. A small minority of cancers are associated with known pathogenic germline variants in high-penetrance cancer-predisposing genes (e.g. *BRCA*, *p53*), but most cancers, including early-onset cancers, develop in the absence of high-penetrance germline mutations[9]. Genome-wide association (GWAS) studies, based on the statistical premise that the same loci are altered more frequently in at least a subset of cases with cancer compared to individuals without cancer, have identified many low-penetrance germline variants that are associated with slightly increased risk of cancer development[10,11]. Currently, about 1,300 cancer risk single-nucleotide polymorphisms (SNPs) have been identified by GWAS studies. However, even when these variants are combined into polygenic risk scores, they only explain a small fraction of heritability of cancer[12].

A limitation of variant-level GWAS analyses is that variants that are not statistically associated with cancer risk could still contribute to cancer development and mediate heritability through phenotypic convergence at biological process level[13]. In phenotypic convergence, distinct genomic alterations in different members of a biological pathway could lead to the same—or similar—phenotypic effect. Germline variants that are not associated with cancer risk could also shape the somatic mutagenesis processes[14] and function as co-oncogenes through interactions with sMs[10,15]. We hypothesize that in all cancers the alterations in biological processes that are required for malignant transformation occur through the combined effects of acquired sMs and common and rare high-functional impact germline polymorphisms (gHFI) in cancer-relevant genes. The gHFI were defined as germline variants that were predicted to alter the structure, expression, or function of protein-coding genes. In this model, germline alterations that contribute to malignant transformation are not necessarily recurrent in the population level to reach statistical significance in association studies and are not sufficiently penetrant to increase cancer risk alone. Instead, such germline abnormalities would increase cancer risk only in the presence of other germline alterations and specific acquired sMs that disable compensatory pathways or activate complementary oncogenic processes. This high level of context-dependent effect implies that the totality of sMs in cancer hallmark genes collectively constitutes driver events in cancer, and the outcome of these sMs is conditional on the constellation of germline high-functional-impact (gHFI) variants in cancer-relevant genes. Based on this hypothesis, we predict that (i) causative germline and somatic variants affect similar sets of genes (those in cancer hallmark pathways) and provide complementary hits; (ii) on average, patients who develop cancer at a younger age will have a higher gHFI variant burden and hence require fewer acquired sMs for malignant transformation than those who develop cancer at older age; and (iii) conversely, sMs play a relatively greater contribution to carcinogenesis in older patients. To test these predictions, we calculate the gHFI variant burden in protein-coding regions of hallmarks of cancer genes in three separate data sets—The Cancer Genome Atlas (TCGA), the Pancancer Analysis of Whole Genomes (PCAWG), and the UK Biobank (UKBB)—and correlate these variant burdens with age at cancer diagnosis. We also examine the gene-level overlap in gHFI and sM, and using TCGA data, assess the correlation between these two different sources of functional variants as a function of age at cancer diagnosis.

## Results

**Germline variants and sMs in the TCGA**. The high-functional-impact sMs were defined as acquired sMs including missense, nonsense, frameshifting, or splice-site altering single-nucleotide changes or indels. In the TCGA, we restricted our analysis to 7468 cancers in patients with European ancestry (to minimize ancestry-related confounders in the germline) corresponding to 31 cancer types. (Supplementary Table 1). Cancers on average harbored 136 gHFI variants (range 79–239) and 216 high-functional impact sM in protein-coding genes (range from 1 mutation in some testicular germ-cell tumors, low-grade gliomas, mesotheliomas, thyroid carcinomas (THCAs), and urothelial bladder carcinomas to 31,087 mutations in a cutaneous melanoma, Fig. 1a). Germline and somatic events tended to affect non-overlapping loci (Fig. 1b). Not all protein-coding genes, when altered, play an equally important role on malignant transformation. Several dozens of oncogenes and tumor suppressor genes have been extensively studied for their biological function, but a much broader range of genes also contribute to cancer biology[16]. The exact number of human genes that enable malignant transformation is unknown and different genes may contribute to a different extent to transformation in different cell types. In this analysis, we use a manually curated list of 1558 cancer hallmark genes assigned to 21 pathways (Supplementary Data 1) and assembled by NanoString Technologies (Seattle, WA). Among these cancer hallmark genes, 1544 genes were affected by sMs, and 1257 were affected in the germline of at least one case across all cancers (Fig. 1c). Figure 1c also illustrates that the majority of cancer hallmark genes ($n = 1253$) can be affected at either germline or somatic level. Cancers on average harbored 13 gHFI variants (range 4–29) and 23 high-functional-impact sM (range 0–3593) in cancer hallmark genes. It is important to note that only 0.38% of gHFI variants in the cancer hallmark genes that we detected have previously been linked to cancer risk in the ClinVar database.

**Age at cancer diagnosis and germline variant burden**. We hypothesize that patients who develop cancer at younger age will carry a greater load of gHFI in cancer hallmark genes, rendering cell lineages more vulnerable to cancer. Therefore, in these individuals fewer acquired sMs are required to trigger malignant transformation. Since there is no uniform age threshold to define early-onset versus late-onset cancers, and the proposed biological phenomenon operates along the continuum of age, we binned TCGA patients into non-overlapping age intervals in increments of 5 years with special two additional groups for the very young (i.e. ≤30) and old (≥81 years). The number of patients in the different age interval groups are not equal, due to increasing cancer incidence with age (Supplementary Fig. 1a). We created the age interval bins to minimize the impact of uneven sample sizes of the various age groups when assessing the age versus variant burden relationship. Cancer types are also not evenly represented across the different age groups: low-grade glioma, thymoma, testicular germ-cell tumors, and pheochromocytoma

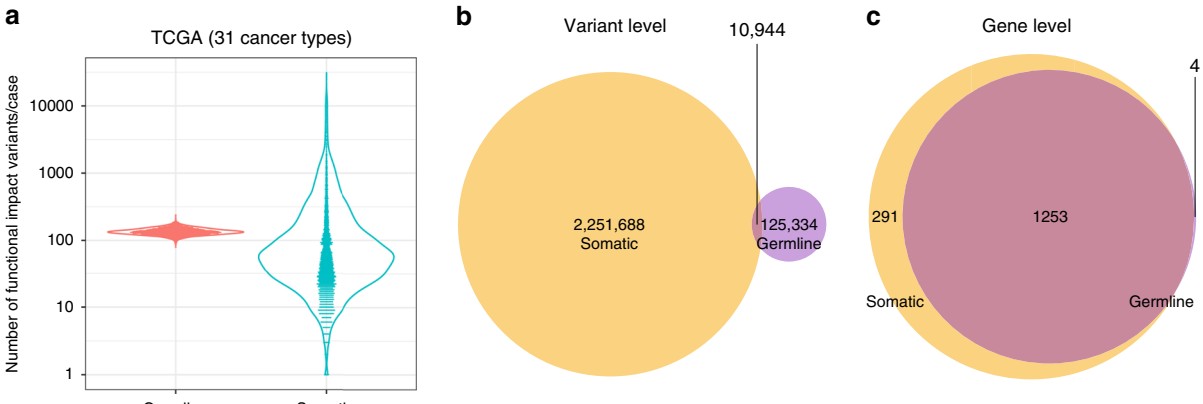

**Fig. 1 Distribution of gHFI variants and somatic mutations in protein-coding genes in the TCGA. a** The number of high-functional impact germline variants and somatic mutations from 7468 patients with 31 different cancer types. **b** Venn diagram of loci affected by germline variants and somatic mutations. **c** Venn diagram of cancer hallmark genes ($n = 1558$) affected by germline variants and somatic mutations in the TCGA population.

are more common in age groups less than 45. Mean age for these cancers was 44 years. In contrast, urothelial bladder carcinoma, lung cancer, and uterine corpus endometrial carcinoma were more common in the age groups older than 65. Mean age for these cancers was 66 years (Supplementary Fig. 1b).

We first tested the association between average $\log_2$-transformed gHFI burden (expressed as number of variants per sequenced megabase) in all protein-coding genes ($n = 19,581$) across the age groups. We found no significant correlation between age and gHFI variant burden for all genes (Fig. 2a, Pearson's $r = -0.21$, $P = 0.54$). Next, we repeated the analysis focusing only on cancer hallmark genes ($n = 1558$), and observed a significant negative correlation (Fig. 2c, $r = -0.77$, $P = 0.0051$) with decreasing gHFI burden as age at diagnosis increased. This higher gHFI burden in younger patients could be driven by the higher incidence of known high-penetrance cancer-predisposition genes in this population. To examine this possibility, we removed all known clinically validated germline cancer predisposition genes from our gene list (Supplementary Data 2). Germline cancer predisposition genes were taken from the National Comprehensive Cancer Network (NCCN) high-risk cancer germline screening guidelines. We repeated the analysis with 1508 genes that remained after removing the known cancer-risk genes. The gHFI variant burden remained significantly and negatively correlated with age of diagnosis (Supplementary Fig. 3a, $r = -0.77$, $P = 0.0058$), indicating that the association is not driven by known cancer-predisposition genes.

When we examined the relationship between age groups and average sM burden, we observed a significant positive correlation between increasing age and higher sM burden for all human genes (Fig. 2b, $r = 0.91$, $P = 0.000091$) and for cancer hallmark genes (Fig. 2d, $r = 0.92$, $P = 0.000073$). As expected, the gHFI and sM burdens showed a strong negative correlation across age groups ($r = -0.70$, $P = 0.017$; Fig. 2e). We estimated the impact of age on this negative correlation in a linear regression model (sM ~ gHFI + Age). The sM burden showed a significant positive correlation with age (Beta $= -0.018$, $P = 0.0030$) but the gHFI variant burden did not (Beta $= 0.46$, $P = 0.91$). This indicates that the strong negative correlation between gHFI and sM across age groups is primarily driven by the age associated increase in sM burden.

To confirm the negative association between gHFI variant burden and age at cancer diagnosis, we examined two independent data sets where germline whole-exome sequence data was available. In the PCAWG ($n = 1487$ cancers), we also observed statistically significant negative correlation between average gHFI variant burden and age at diagnosis as seen in the

TCGA (Fig. 3a, $r = -0.61$, $P = 0.047$). In the UKBB data ($n = 7198$ cancers), we found a negative correlation between gHFI and age, but the trend did not reach the threshold for statistical significance (Fig. 3b, $r = -0.55$, $P = 0.16$).

Because the underlying principle applies across cancer types and the combined data provides greater power to detect associations, we performed these analyses on data combining all cancers together. However, in the TCGA data, we could also examine the correlation between average gHFI and sM burdens across age groups within each of the 23 cancer types that included at least 100 cases of European ancestry separately. The sample sizes for the cancer cohorts ranged from 109 to 745, and the sample sizes of the age subgroups were substantially smaller, leading to variable and low power to detect statistically significant correlations by cancer type. In most cancer types we observed a negative trend between gHFI burden and age except in glioblastoma, urothelial bladder carcinoma, and stomach adenocarcinoma. The negative association reached nominal statistical significance (unadjusted for multiple comparisons) in ovarian cancer, thyroid carcinoma, and papillary renal cell carcinoma (Supplementary Fig. 4). In pheochromocytoma, ovarian, liver, glioblastoma, and thymoma, we also observed a significant negative correlation between gHFI and sM burdens across age groups. Unexpectedly in uterine cancer, lung squamous cell cancers, and pancreatic cancer, we observed the opposite relationship increasing gHFI and increasing sM burden in older patients (Supplementary Fig. 5). Whether this difference in relationship indicates distinct oncogenic processes in these tissues, or represents statistical anomalies due to modest sample sizes, will need to be investigated.

We also examined the correlations between age and gHFI and sM burdens as continuous variables across all patients in the TCGA cohort. We again observed a weak negative but highly significant correlation between age and gHFI burden ($r = -0.031$, $P = 0.0070$). The positive correlation between age and sM burden stronger and also statistically highly significant ($r = 0.24$, $P < 2.45 \times 10^{-95}$, Supplementary Fig. 6). These findings suggest that there are powerful confounders that weaken the association between age at cancer diagnosis and deleterious germline variant burden or sM burden that particularly affect cancers that develop in ages between 50 and 70. To further assess interaction between age, sM, and gHFI as continuous variables at individual patient level we performed linear regression analysis to adjust for the effect of uneven cancer types distribution in each age interval. After adjusting for cancer type, the age of diagnosis remained significantly correlated with gHFI (Beta $= -0.00023$, $P = 0.032$) and sM (Beta $= 0.0025$, $P = 6.79 \times 10^{-12}$) burdens.

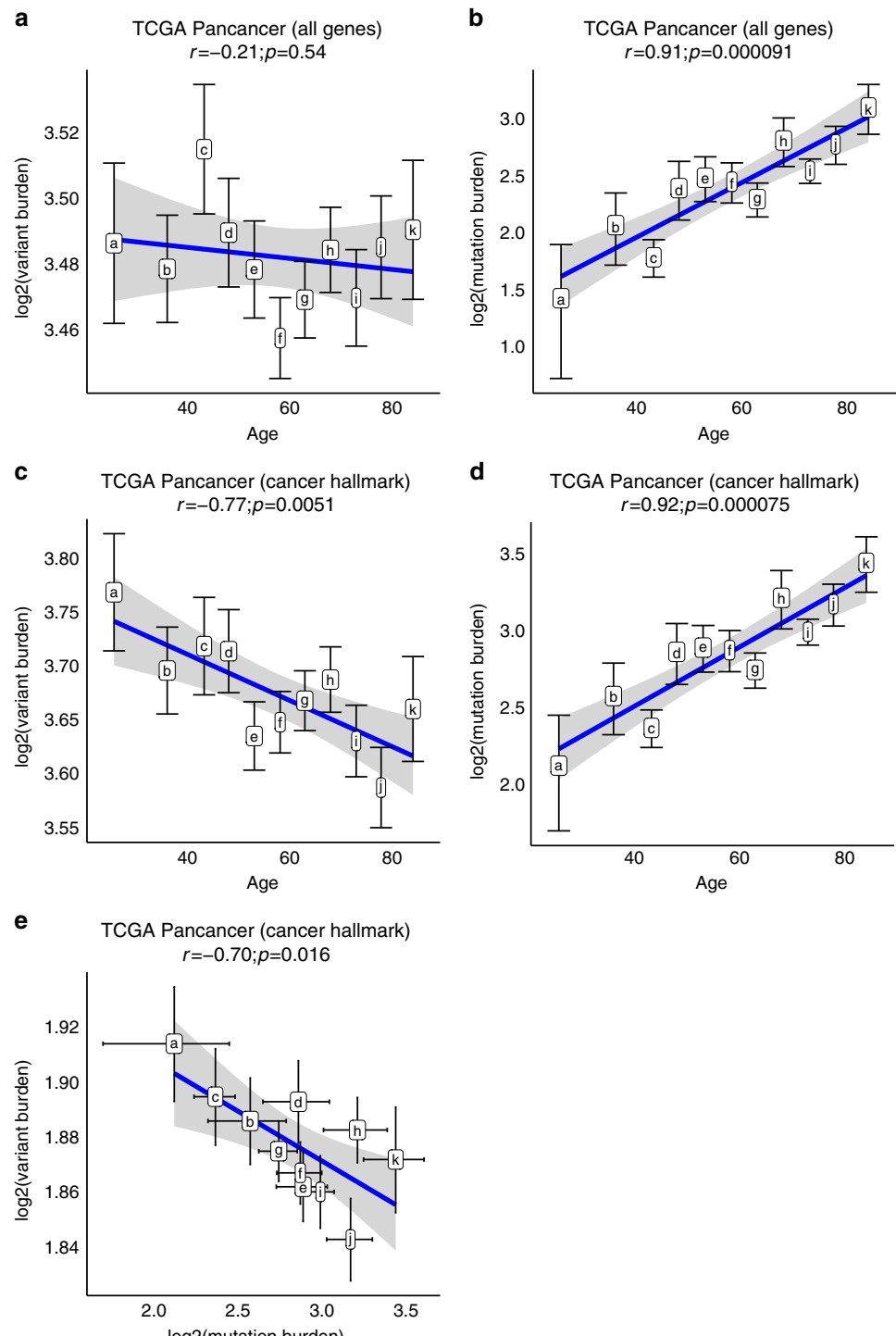

**Fig. 2 Correlations between the average gHFI variant burden and somatic mutation burden and age in the TCGA ($n = 7468$ cases). a** Variant burden in all human genes ($n = 19,581$ genes) versus age intervals. **b** Mutation burden in all human genes versus age intervals. **c** Variant burden in cancer hallmark genes ($n = 1558$ genes) versus age intervals. **d** Mutation burden in cancer hallmark genes versus by age intervals. **e** Correlation between variant burden and mutation burden in cancer hallmark genes across age. Tags a–k indicate age intervals corresponding to ages ≤30 ($n = 307$), 31–40 ($n = 545$), 41–45 ($n = 405$), 46–50 ($n = 568$), 51–55 ($n = 820$), 56–60 ($n = 1014$), 61–65 ($n = 1080$), 66–70 ($n = 958$), 71–75 ($n = 820$), 76–80 ($n = 572$), and ≥81 ($n = 379$) years. The y-axes correspond to log$_2$ transformed variant/mutation burden. Error bars represent standard error. The $r$ represents Pearson correlation coefficient. Spearman's Rho test (two-sided) was used to generate the $p$ value to measure the strength of correlation coefficient.

**Cancer pathways are affected by germline and somatic events**. Next, we examined the average fraction of altered genes affected by either gHFI or by sM in each of the 21 cancer hallmark pathways in the TCGA (Fig. 4a). All pathways were affected by both types of events although the contribution of gHFI versus sM

was highly variable across pathways (Fig. 4b). Germline variants were the dominant source of effect on innate immunity, epithelial–mesenchymal transition, and extracellular matrix pathways whereas sMs were the dominant contributor to alterations in cancer driver genes, cell cycle and apoptosis, WNT and

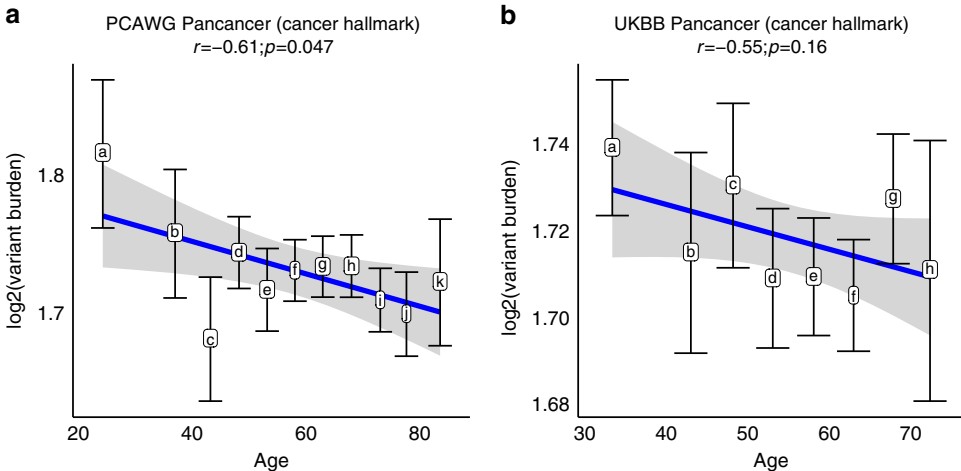

**Fig. 3 Correlation between gHFI variant burden in cancer genes in the PCWAG and UKBB. a** Variant burden for 11 age intervals in the PCAWG ($n = 1487$ cases, the tags a–k correspond to ages ≤30, 31–40, 41–45, 46–50, 51–55, 56–60, 61–65, 66–70, 71–75, 76–80, and ≥81). **b** Variant burden in eight age intervals in the UKBB ($n = 7198$ cases, the tags a–h correspond to ages ≤40, 41–45, 46–50, 51–55, 56–60, 61–65, 66–70, and ≥71). The y-axes show the $\log_2$ transformed variant/mutation burden. Error bars represent standard error. The $r$ represents Pearson correlation coefficient. Spearman's Rho test (two-sided) was used to generate the p-value to measure the strength of correlation coefficient.

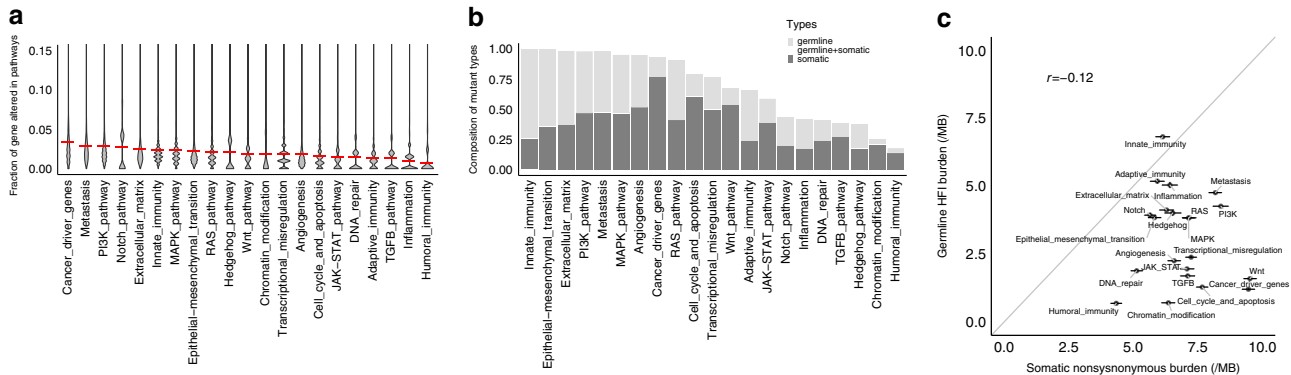

**Fig. 4 Contribution of germline variants and somatic mutations to alterations in cancer pathways. a** Violin plots show the average fraction and distribution of member genes affected by either germline variants or somatic mutations in 21 pathways. Red lines indicate the average fraction of affected genes. **b** Average percent of the contribution of germline variant versus somatic mutation to all the affected genes in each pathway across all TCGA samples. **c** Correlation between average germline variant burden and somatic mutation burden across the cancer hallmark pathways. Error bars show the standard error. Vertical error bars (for the germline HFI burden/MB) are too small to be discernable.

TGFbeta pathways. Figure 4c shows the correlation between average germline variant burden and average sM burden for the 21 pathways across all cancer types.

We further evaluated the average fraction of gHFI and sM in the 21 cancer hallmark pathways at cancer-type level including the 23 cancer types with at least 100 cases. The contribution of gHFI variant and sM to the total number of pathway alterations varied by cancer type. sMs were the dominant source of alterations in cutaneous melanoma, lung squamous cell carcinoma, uterine corpus endometrial carcinoma, urothelial bladder carcinoma, and colon adenocarcinoma. Notably, most of these cancers are strongly associated with environmental carcinogen exposure. In contrast, gHFI variants showed stronger contribution to testicular germ-cell tumor, pheochromocytoma and paraganglioma, thyroid carcinoma, and prostate cancer (Supplementary Fig. 7).

## Discussions

Cancer biology tends to focus on genes with frequent sMs that have a large impact on cell proliferation and cell viability. Many

of these genes were discovered three or four decades ago as transforming oncogenes and tumor suppressor genes[17]. In contemporary literature, alterations in these genes are often described as cancer-driver events and the large number of other, seemingly random sMs in a tumor are considered passenger mutations with little importance for cancer biology[18]. However, many passenger mutations are predicted to alter protein function and affect genes that could be important in bringing about the malignant phenotype[19]. All sMs accumulate against the background of the unique combination of germline variants in protein-coding genes, and in non-coding regulatory sequences that each individual harbors[10,14]. Mounting evidence suggests that this genomic (and epigenomic) context influences the biological impact of sMs in classical oncogenes and tumor suppressor genes[10,11]. The cell-line-restricted effect of transforming oncogenes was recognized since their discovery. Transgenic mouse models also demonstrate mouse strain-specific effects. A growing list of common germline variants in cancer-relevant genes have been experimentally demonstrated to impact protein function without having a detectable association with cancer risk[15]. For example, the M326I variant (rs3730089, variant allele frequency [VAF] 22%) in the

p85α regulatory subunit of phosphatidylinositol 3-kinase (*PI3K*) results in constitutively increased activity of the PI3K pathway[20]. These data support the hypothesis that the combined effect of sMs and many germline variants, including common and rare variants that are not associated with increased cancer risk, lead to malignant transformation.

Our analysis has demonstrated that genes mediating the important biological processes in cancer are affected by both germline variants and sMs and both types of alterations are present in large numbers in each cancer. This dual appearance suggests a functional complementarity that is consistent with a growing number of functional studies in the laboratory that demonstrate functional interaction between germline and somatic events. For example, the common human germline variant rs56391007 (T1010I, VAF 1%) in the hepatocyte growth factor receptor (*cMET*) increases colony formation, cell migration, and invasion when introduced into MCF-10A immortalized breast epithelial cell line that already harbors heterozygous deletion of the *CDKN2A* locus and amplification of *MYC* along with other somatic alterations in *BRAF*, *EGFR*, *PIK3CA*, *ERBB2*, and *ALK* genes[21]. Another example is germline deletion of *PALB2* in transgenic mice that leads to mammary tumor formation with a long latency—however, co-deletion of *TP53* accelerates tumor formation dramatically, providing support for synergistic interactions between a germline deletion and a frequently observed sM in a mouse model system[22]. Correspondingly, we have shown in three independent data sets that patients who develop cancer at younger age have on average a greater load of high-functional-impact germline variants in cancer hallmark genes than do individuals who develop cancer at an older age. This differential mutational load is consistent with the hypothesis that some individuals reach a critical level of alterations in cancer hallmark genes that are required for carcinogenesis with fewer sMs because they carry more deleterious germline variants. Indeed, we observed a statistically significant inverse relationship between gHFI variant burden and sM burden in cancer hallmark genes across age groups in the TCGA. However, at the level of individual cancer types, and at patient level, substantial heterogeneity of the results was observed. In most cancer types, when examined separately, the negative trend between gHFI variants and sM across age intervals was weak; it met our threshold for statistical significance only in five cancer types. The power of cancer-type-specific analyses is limited by sample size, but the weak correlations also suggest that many other factors can also influence at what age cancer develops in an individual.

Environmental exposure to carcinogens and mutations in DNA repair genes can alter the rate of accumulation of sMs and distort a simple age-related negative relationship between gHFI variant and sM burden. Single-nucleotide changes and indels do not fully capture the extensive genomic changes in cancers that include deletions and amplifications of otherwise structurally intact genes as well as epigenetic changes that each can influence gene expression levels. Cancer cell proliferation rate can influence the duration of the latent phase of cancer before it becomes clinically detectable, some cancers, particularly those that develop from premalignant lesions, likely have been growing and accumulating alterations for many years, maybe even decades, before diagnosis. For sMs, we observed an increase in mutation number and mutation burden with increasing age in both all human genes and the subset that are cancer hallmark genes. This is consistent with reports that demonstrated accumulation of mutations in normal tissues during aging[23].

Testicular germ-cell tumors (TGCT), thyroid carcinoma (THCA), and low-grade glioma (LGG) were unique in the TCGA data because these cancers dominated the youngest age groups (253 cases had one of these 3 cancers out of 545 patients under age 40). The early onset of these cancers is well known and family

and twin studies showed high heritability although no shared high risk germline alterations were identified that could explain it. For example, heritability of TGCTs is between 37% and 49% which is higher than breast (31%) or ovarian cancers (39%)[24,25], but known GWAS TGCT-associated germline risk variants only explain about 9% of TGCT heritability[25]. Similarly, about 20–25% of THCAs are familial[26] and the heritability of gliomas is about 25%[25]. There are no know recurrent, high-penetrance germline variants associated with these cancers[9]. Our results suggest that the combined effects of deleterious germline variants in cancer biology-related genes may account for at least some of the heritability.

This study cannot answer what, and how many genes must be affected by sMs or deleterious germline variants before malignant transformation is complete, or how germline variants and somatic events complement each other at the level of relevant pathways. Gene memberships in pathways are fluid concepts, depending on the organizing principles and selection criteria adopted by investigators. We also assumed that our list of 1558 genes encompasses most genes that are important for cancer biology, and in our calculations we assigned equal importance to each of these in the different cancer types. These assumptions are likely simplifications, different genes may have variable importance in different cancer types and the number of genes that could influence cancer biology is likely larger than 1558. However, despite these limitations, in three data sets, each including thousands of different solid tumors we found that patients who develop cancer at a younger age carry on average a greater number of deleterious germline variants in a manually curated list of 1558 cancer related genes than patients who develop cancer at an older age. We note that very few (~0.38%) of these germline protein-coding variants have been linked to cancer risk. In the TCGA, where both germline and sM data were available, we also observed a significant negative correlation between deleterious germline variant burden and average sM burden in the same 1558 cancer-related genes across the increasing age groups. Overall, these results suggest that in early-onset cancers, the larger number of germline variants in cancer genes implies a greater contribution of germline alterations to the transformation process; in contrast, late-onset cancers are more dependent on acquired sMs.

## Methods

**Study subjects.** Germline variants of 10,389 patients included in the TCGA corresponding to 33 cancer types generated by Huang et al. were downloaded from the Genomic Data Commons (GDC, https://gdc.cancer.gov/about-data/publications/PanCanAtlas-Germline-AWG) of the National Cancer Institute (NCI)[9,27]. The sMs were obtained from the TCGA PanCancer Altlas MC3 set[28]. We restricted our analysis to patients of European ancestry, the largest ancestry cohort, to maximize statistical power while accounting for potential population stratification effects. We only kept individuals with data on both germline and sMs. In addition, we restricted our analysis to solid tumor. We also excluded pediatric patients with a cancer diagnosis before age 18. Our final analysis included 7468 cancer patients and 31 cancer types. We included variants of autosomal exome region that met the quality control metrics set by Huang et al.[9], variants with at least five counts of the alternative allele and alternative allele frequency of at least 30%. The included samples and patient characteristics are summarized in Supplementary Table 1.

Germline variants of 1823 PCAWG patients were downloaded from the ICGC Data Portal (https://dcc.icgc.org/)[14]. We excluded 131 samples without age of diagnosis information and an additional 205 patients with cancer diagnosis before age 18. The final analysis set included 1483 cases; none of these patients were included in the TCGA.

Germline variants derived by exome sequencing of 49,960 individuals were obtained from UK Biobank (UKBB) including 8,959,608 variants[29]. The germline data generated according to the Functionally Equivalent analysis pipeline[30] and filtered with inbreeding coefficient <−0.03 or without at least one variant genotype of DP ≥ 10, GQ ≥ 20 and, if heterozygous, AB ≥ 0.20. Our analysis only focuses on the 8,758,489 autosome variants (http://biobank.ctsu.ox.ac.uk/showcase/label.cgi?id=170). In addition, based on the genetic ethnic information provided by UKBB, only 41,212 (82.4%) samples with European ancestry were included in this study.

We further selected the variants in the exome region of 19,396 genes and discarded variants with a missing rate larger than 20% across all the individuals. The final variant set included 3,965,725 high-quality variants. We identified individuals with cancer ($n = 7205$, age range 11–76 years) as described by Cristopher et al.[29], data available at UKBB. We further excluded pediatric patients with diagnosis before age 18, leaving 7198 cases in our final analysis. This work was conducted under UK Biobank application 29900.

The sequencing protocols and variant calling pipelines are different between TCGA, PCAWG, and UKBB. The average exome-sequencing depth of TCGA normal sample is approximately 100×. The PCAWG data in our analysis are derived from whole-genome sequencing with the mean read depth equal to 39× in normal samples. The depth of UKBB exome-sequencing data exceeds 20× at 94.6% of exome capture sites on average. TCGA and PCAWG pancancer consortium used the same pipeline for germline variants calling[9], but UKBB used a different Functionally Equivalent (FE) pipeline.

**High-functional impact variants.** The functional impact of missense germline variants was predicted using MetaSVM ensemble algorithm[28] and annotation by ClinVar database when available[31]. We considered a missense variant high-functional impact if classified as Deleterious by MetaSVM or listed as Pathogenic/Likely-Pathogenic in ClinVar. Variants annotated as Tolerance by MetaSVM or Benign in ClinVar were excluded. We used MetaSVM scores from the dbNSFP database which includes pre-calculated function impact scores for 75,931,005 human non-synonymous single-nucleotide variants[32]. Loss-of-function (LoF) variants, including frameshift indels, stop gain, and stop loss variants, were also considered high-functional impact as well as variants annotated as high-confidence LoF in gnomAD[33] or Pathogenic/Likely-Pathogenic in ClinVar. Variants annotated as Benign in ClinVar were excluded. The variant burden in a particular sample was calculated as the total number of variants in the regions of selected genes divided by total length of the selected genes in megabases.

**Statistical analyses.** The TCGA and PCAWG samples were separated into eleven age groups based on patient age at diagnosis (≤30, 31–40, 41–45, 46–50, 51–55, 56–60, 61–65, 66–70, 71–75, 76–80, ≥81). In the UKBB, very few individuals had cancer between ages 18–30 years or after age 70; therefore, in this data set we created only eight age groups (≤40, 41–45, 46–50, 51–55, 56–60, 61–65, 66–70, ≥71). We calculated the average variant burden of gHFI and sM for each group along with the standard error of the mean. The Pearson correlation coefficient was used to assess the relationship between gHFI and sM burden across increasing age groups. Spearman's Rho test was employed to calculate statistical significance and $P \leq 0.05$ was considered significant. All analyses were performed in R (http://www.R-project.org/).

**Reporting summary.** Further information on research design is available in the Nature Research Reporting Summary linked to this article.

## Data availability

The TCGA germline variants are available at https://gdc.cancer.gov/about-data/publications/PanCanAtlas-Germline-AWG. The TCGA somatic mutations are available at https://gdc.cancer.gov/about-data/publications/mc3-2017. The germline PCAWG data set is available at https://icgc.org/daco. Access to the germline UKBB data can be requested at https://www.ukbiobank.ac.uk/. Data supporting the findings of this study are available in the Article, Supplementary Information, or from the authors upon reasonable request.

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

## Acknowledgements

We thank Laurene Goode for providing clerical assistance to coordinate this project. Research reported in this publication was supported by grants from the Breast Cancer Research Foundation and the Susan Komen Foundation (SAC160076) to L.P.

## Author contributions

L.P. and T.Q. designed the study. T.Q. conducted most of the analyses. H.M. prepared the germline data of PCAWG. Y.Y and H.Z prepared the germline data of UKBB. L.P. and T.Q. wrote the article. M.M., T.O.'M., H.Z., J.T., M.G., C.H., and Y.K. contributed to data interpretation and manuscript writing.

## Competing interests

L.P. has received honoraria from Astra Zeneca, Merck, Novartis, Bristol-Myers Squibb Genentech, Eisai, Pieris, Immunomedics, Seattle Genetics, Clovis, Syndax, H3Bio, and Daiichi. The remaining authors have no conflict of interests to declare.
