## [Peer Review File · Nature Communications]

Reviewers' comments:

Reviewer #1 (Remarks to the Author):

The paper by Qing et al. would be an important advance if true. It is likely that the observed effects are caused by uncontrolled and understood variables.

0) The number one concern is that the report provided represents our knowledge of inherited mendelian cancer syndromes. The authors should remove all strong germline defects in all genes known to cause mendelian syndromes and show that the effect remains. The list includes, but is not limited to, Lynch, HBOC, Li-Fraumeni, von Hippel Lindau, etc..

1) A more insightful way to plot 2C might be excess deleterious variants vs median age, or some such. It is not immediately clear that there is 1/2 extra variant in the cancer hallmark genes in the youngest group than the oldest group.

2) Figure 2E needs to regress age from association between deleterious variants and mutation burden. The literature that age is directly associated to mutation number in normal tissue is extremely strong and the correlation between age and mutation number is better than between germline and somatic mutations.

3) Figure 3A and 3B, why is the number of deleterious variants so much lower than in 2C? The number of the genes should be similar and so the effect should be similar.

4) Given positive selection in innate immunity genes for variation it would be stunning if the pathway effect were not observed there.

5) More complicated is the fact that the distribution of cases vs age is non-uniform. If LGG had common inherited drivers at all ages, it would create the distribution presented as it represents such a large fraction of young cases. It would be Useful to regress out the age effects (if possible) of tumor type effect.

Reviewer #2 (Remarks to the Author):

Qing et al evaluate the associated characteristics of germline high functional impact variants (gHFI) and somatic mutations:

(sM) in regard of age of cancer diagnosis. Analysing TCGA, ICGC and Biobank, they assert that:

(1) gHFI and sM affect overlapping genes

(2) the average number of gHFI in cancer hallmark genes is higher in patients who develop cancer at a younger age

(3) the average number of sM increases in increasing age groups

(4) average gHFI and average sM burden are negatively correlated in increasing age groups

Overall

This analysis does not seem especially novel either in the questions it asks or in the findings generated.

It is long established that cancers associated with germline susceptibility typically arise at a younger age. Burden analyses in known and candidate CSGs have shown higher burden of rare variants in 'enriched' cancer subjects (young onset and/or familial).

It is also well established that embryonal tumours are quiet and that for many cancers related to environmental mutagens, the observed TMB will correlate with age of diagnosis.

I am not sure that this analysis adds much to the debate, aside from showing the above to be true via negative correlation at pretty unimpressive levels of effect size/statistical significance having utilised an arbitrarily selected (and "manually curated") set of cancer-associated genes.

Major Points

1) My major concern regards multiple testing. Both UKBiobank and TCGA represent very large dataset well-powered to generate very significant p-values in the face of biologically important associations. Results presented are of pretty marginal significance (eg $p=0.047$ p4 L157) and one is unclear the total denominator of testing they reflect. More explicit presentation is required of the total number of statistical tests with due adjustment for multiple testing.

Minor Points

1) I find intellectually frustrating use of the term 'variant' to mean germline and 'mutation' to mean somatic. They are all either variants or mutations. However, I concede that the notation used does makes this distinction clearer

Reviewer #1

General comment:

"The number one concern is that the report provided represents our knowledge of inherited mendelian cancer syndromes. The authors should remove all strong germline defects in all genes known to cause mendelian syndromes and show that the effect remains. The list includes, but is not limited to, Lynch, HBOC, Li Framueni, von Hippel Lindau, etc..."

This is an important comment and we have repeated our analysis excluding all known clinically validated cancer-predisposition genes from our cancer hallmark genes. We took the list of germline cancer predisposition genes from the National Comprehensive Cancer Network (NCCN) high risk cancer germline screening guidelines. Fifty genes in our hallmarks of cancer gene list were associated with cancer risk and were removed; these 50 genes are listed in a new Supplementary Table 3. Results of the repeat analysis are included as a new Supplementary Figure 2. The associations and significance values have not changed after removing these genes and therefore we included these results as supplementary figures, rather than replacing the main figures. To describe these new results, we inserted the following section on page 4.

"This higher gHFI burden in younger patients could be driven by the higher incidence of known high penetrance cancer predisposition genes in this population. To examine this possibility, we removed all known clinically validated germline cancer predisposition genes that from our gene list (Supplementary Table 3). Germline cancer predisposition genes were taken from the National Comprehensive Cancer Network (NCCN) high risk cancer germline screening guidelines. We repeated the analysis with 1508 genes that remained after removing the known cancer risk genes. The gHFI variant burden remained significantly and negatively correlated with age of diagnosis (Supplementary Figure 2a, $r = -0.77$, $P = 0.0058$) indicating that the association is not driven by known cancer predisposition genes."

"We repeated the analysis with 1508 genes without the known cancer risk genes. The gHFI variant burden remained negatively correlated with age of diagnosis in PCAWG (Supplementary Figure 2b, $r = -0.65$, $P = 0.031$) and UKBB (Supplementary Figure 2c, $r = -0.55$, $P = 0.16$) cohorts."

Specific comments:

1) A more insightful way to plot 2C might be excess deleterious variants vs median age, or some such. It is not immediately clear that there is 1/2 extra variant in the cancer hallmark genes in the youngest group than the oldest group.

We plotted figure 2C to keep it consistent with all other plots that show log₂ transformed absolute variant burden by age groups. These plots directly illustrate the trend of average variant burden across increasing age groups. As suggested by the reviewer, we also plotted relative (i.e. excess) deleterious variant burden compared to the oldest age group as the reference population. This plot also supports our conclusion of greater germline variant burden in cancer related genes younger patients as seen below. The y-axis shows log₂ transformed variant burden rate relative to the average burden in the oldest age cohort k.

2) Figure 2E needs to regress age from association between deleterious variants and mutation burden. The literature that age is directly associated to mutation number in normal tissue is extremely strong and the correlation between age and mutation number is better than between germline and somatic mutations.

The reviewer is correct the inverse relationship is primarily driven by the increasing somatic mutation burden with age. We applied a regression model including germline variant burden, somatic mutation burden and age. The results are now included on page 4 as:

“We estimated the impact of age on the negative correlation between gHFI and sM burdens in a linear regression model ($sM \sim gHFI + Age$). The sM burden showed a significant positive correlation with age ($Beta = - 0.018, P = 0.0030$) but the gHFI variant burden did not ($Beta = 0.46, P = 0.91$). This indicates that the strong negative correlation between gHFI and sM across age groups is primarily driven by the age associated increase in sM burden.”

3) Figure 3A and 3B, why is the number of deleterious variants so much lower than in 2C? The number of the genes should be similar and so the effect should be similar.

We thank the reviewer for pointing out this inconsistency. The Y-axes in Figure 2A, C and E were showing average log2 transformed variant count not variant burden (which is variants per sequenced megabase) as shown in Figure 3A and 3B. We harmonized the figures and the revised Figures 2A, C and E now show log2 transformed variant burdens.

4) Given positive selection in innate immunity genes for variation it would be stunning if the pathway effect were not observed there.

We agree with the reviewer. We take this as indication of the integrity of the data sets. We kept the immune pathway in the figures for sake of completeness.

5) More complicated is the fact that the distribution of cases vs age is non-uniform. If LGG had common inherited drivers at all ages, it would create the distribution presented as it

represents such a large fraction of young cases. It would be useful to regress out the age effects (if possible) of tumor type effect.

This is a very important comment and is central to our research question. We know from the clinical literature that most Low Grade Gliomas (LGG), testicular germ cell tumors (TGCT), or thyroid carcinomas (THCA) that dominate the young cancer age groups occur in patients who carry NO KNOWN cancer predisposing germline mutation. There are no common inherited germline causes for these and most other early onset cancers. To address the contribution of the totality of germline deleterious variant burden in cancer hallmark genes in these and all other cancer types we included a new Supplementary Figure 3 that shows the correlation of average gHFI and age groups for each cancer type. We also inserted a description of these results on Page 5:

“In most cancer types we observed a negative trend between gHFI burden and age except in glioblastoma, urothelial bladder carcinoma, stomach adenocarcinoma. The negative association reached nominal statistical significance (unadjusted for multiple comparisons) in ovarian cancer, thyroid carcinoma and papillary renal cell carcinoma (Supplementary Figure 3).”

We also used linear regression to assess the effect of cancer type at the individual level and compared two models: (1) gHFI~Age and (2) gHFI~Age+CancerType. Even after including cancer type, we still observed a significant negative association between gHFI burden and age ($p=0.031$). When we assessed somatic variant burden the same way, we also found that age was independently significant. We include these results in the revised manuscript on page 5:

“To further assess interaction between age, sM and gHFI as continuous variables at individual patient level we performed linear regression analysis to adjust for the effect of uneven cancer types distribution in each age interval. After adjusting for cancer type, the age of diagnosis remained significantly correlated with gHFI ($Beta = -0.00023$, $P = 0.032$) and sM ($Beta = 0.0025$, $P = 6.79 \times 10^{-12}$) burdens.”

In addition, we also expanded the discussion section on page 7 with the following paragraph and included the additional references to support the statements:

“Testicular germ cell tumors (TGCT), thyroid carcinoma (THCA) and low-grade glioma (LGG) were unique in the TCGA data because these cancers dominated the youngest age groups (438 cases had one of these 3 cancers out of 852 patients under age 40). The early onset of these cancers is well known and family and twin studies showed high heritability although no shared high-risk germline alterations were identified that could explain it. For example, heritability of TGCTs is between 37-49% which is higher than breast (31%) or ovarian cancers (39%)^{24,25}, but known GWAS TGCT-associated germline risk variants only explain about 9% of TGCT heritability²⁵. Similarly, about 20-25% of THCAs are familial²⁶ and the heritability of gliomas is about 25%²⁵. There are no known recurrent, high penetrance germline variants associated with these cancers⁹. Our results suggest that the combined effects of deleterious germline variants in cancer biology related genes may account for at least some of the heritability.”

Reviewer #2

Overall comments

This analysis does not seem especially novel either in the questions it asks or in the findings generated. It is long established that cancers associated with germline susceptibility typically arise at a younger age. Burden analyses in known and candidate CSGs have shown higher burden of rare variants in ‘enriched’ cancer subjects (young onset and/or familial). It is also well established that embryonal tumors are quiet and that for many cancers related to environmental mutagens, the observed TMB will correlate with age of diagnosis. I am not sure that this analysis adds much to the debate, aside from showing the above to be true via negative correlation at pretty unimpressive levels of effect size/statistical significance having utilized an arbitrarily selected (and “manually curated”) set of cancer-associated genes.

We agree with the reviewer that our results build on previous concepts. Indeed, the idea that mutations accumulate over time and contribute to aging related diseases is almost a hundred years old. However, technology only allowed to demonstrate this in the past few years, due to dropping costs of large scale sequencing. We have added several new references to acknowledge this. We consider our work an important contribution because the strong age dependent increase in deleterious germline variant in cancer relevant genes and its inverse relationship with somatic mutation burden has not previously been shown on a pan-cancer basis. The cancer field continues to be dominated by search for the few critical cancer drivers. We hope that our paper adds to the growing voice of an alternative view that emphasizes that cancer can arise through a multitude of different mutations and the totality of these determines the onset and behavior of a particular cancer.

We also agree with the reviewer comments about the arbitrariness of our 1558 gene list. We selected these because each has been experimentally shown to have an effect on cancer in at least some model systems and we hoped that such curated hallmark of cancers gene list would be able to demonstrate what we hypothesized. To point out the limitation of selecting a uniform list of “cancer genes” we inserted the following sentences into the discussion section on page 7:

“We also assumed that our list of 1558 genes encompasses most genes that are important for cancer biology, and in our calculations we assigned equal importance to each of these in the different cancer types. These assumptions are likely simplifications, different genes may have variable importance in different cancer types and the number of genes that could influence cancer biology is likely larger than 1558.”

Specific Major Points

1) My major concern regards multiple testing. Both UKBiobank and TCGA represent very large dataset well-powered to generate very significant p-values in the face of biologically important associations. Results presented are of pretty marginal significance (eg $p=0.047$ $p4$ L157) and one is unclear the total denominator of testing they reflect. More explicit presentation is required of the total number of statistical tests with due adjustment for multiple testing.

Our main analysis does not include multiple testing of a large number of different hypotheses. The main figures present results of a pan-cancer analysis that includes all cancers in all 3 data sets. We use the TCGA as a discovery dataset, and the PCWAG and UKBB as a validation dataset, testing the same question, the significance of trend of average germline variant burden across 11 a priori defined age groups in all 3.

The reviewer is correct about multiple hypothesis testing on Supplementary figures 3 and 4, where we show results by cancer type in the TCGA. We did not adjust the values for multiple testing in these figures and this is now clearly stated in the text as well as in the corresponding figure legends.

On page 5 we inserted the following statement:

“In most cancer types we observed a negative trend between gHFI burden and age except in glioblastoma, urothelial bladder carcinoma, stomach adenocarcinoma. The negative association reached nominal statistical significance (unadjusted for multiple comparisons) in ovarian cancer, thyroid carcinoma and papillary renal cell carcinoma (Supplementary Figure 3).”

Also:“Supplementary Figure 3. Correlations between average germline high-functional variant burden versus age of diagnosis by age intervals in each TCGA cancer type. The red boxes highlight the significant associations with $P < 0.05$. **P-values are not adjusted for the multiple group comparisons.** The tags a-k correspond to ages ≤ 30 , 31-40, 41-45, 46-50, 51-55, 56-60, 61-65, 66-70, 71-75, 76-80, and ≥ 81 .”

Supplementary Figure 4. Log-log age interval scatter plots between average germline high-functional variant burden versus the mutation burden by age intervals in each TCGA cancer type. The red boxes highlight the significant associations with $P < 0.05$. **P-values are not adjusted for the multiple group comparisons.** The tags a-k correspond to ages ≤ 30 , 31-40, 41-45, 46-50, 51-55, 56-60, 61-65, 66-70, 71-75, 76-80, and ≥ 81 .”

Minor Point

I find intellectually frustrating use of the term ‘variant’ to mean germline and ‘mutation’ to mean somatic. They are all either variants or mutations. However, I concede that the notation used does makes this distinction clearer

We understand the concern of the reviewer and view it the same way. A change in a nucleic acid sequence is what matters and it is odd to call the change differently in different contexts. We try to use the terminology in the conventional way referring to a nucleotide change as mutation when it occurs in cancer and as a variant when it is seen in the normal tissue.

The reviewer understood our intention clearly, using this conventional terminology we try to distinguish DNA sequence changes by origin, acquired versus inborn.

REVIEWERS' COMMENTS:

Reviewer #1 (Remarks to the Author):

Given that the effects of age and mutation rate are driven by age and are well reported there is no reason to include panels 2b, 2d, and 2e.

There is still a data discrepancy. Log2 variant burden is shown in panels 2a and 2c for the same sets of samples yes, moving from all genes to cancer hallmarks actually increased the number of variants (~ 1.8 to ~ 1.88).

There is also an unexplained data discrepancy in PCAWG, where the number of log2 number of variants is reported to be ~ 1.75 and then even lower to ~ 1.72 in the UKBB dataset. Why should there be different number of deleterious variants in these populations?

Yet more oddness is that the number of mutations seems to have gone up with removal of the cancer risk genes.

I don't think Figure 4 adds much to the story. I have read it and think it means nothing.

The supplements should be done without known driver genes. Ovarian should go away.

Supp4 all results should be FDR corrected.

Reviewer #2 (Remarks to the Author):

The manuscript has certainly been improved with the addition of the (i) repeat analysis with removal of established cancer susceptibility genes, and (ii) stratification of associations by tumour type.

The authors have inserted some covering comments into the text in response to my comments regarding (i) use of a somewhat arbitrary gene list (ii) multiple testing, which, whilst not addressing the issues per se, at least confer transparency.

I overall remained concerned about lack of novelty regarding hypotheses/observations, as well as the rather marginal p-values relating to gHFI and age.

Note: from current GWAS, >35% of the heritability has been explained. Heritability analyses demonstrate that the vast majority of TGCT heritability resides in common variants tagged by a standard GWAS array. (Litchfield, Thomsen et al. 2015, Litchfield, Levy et al. 2017, Wang, McGlynn et al. 2017)

Note: there are numerous references to gremlin variants!

Litchfield, K., M. Levy, G. Orlando, C. Loveday, P. J. Law, G. Migliorini, A. Holroyd, P. Broderick, R. Karlsson, T. B. Haugen, W. Kristiansen, J. Nsengimana, K. Fenwick, I. Assiotis, Z. Kote-Jarai, A. M. Dunning, K. Muir, J. Peto, R. Eeles, D. F. Easton, D. Dudakia, N. Orr, N. Pashayan, D. T. Bishop, A. Reid, R. A. Huddart, J. Shipley, T. Grotmol, F. Wiklund, R. S. Houlston and C. Turnbull (2017).

"Identification of 19 new risk loci and potential regulatory mechanisms influencing susceptibility to testicular germ cell tumor." *Nat Genet* 49(7): 1133-1140.

Litchfield, K., H. Thomsen, J. S. Mitchell, J. Sundquist, R. S. Houlston, K. Hemminki and C. Turnbull (2015). "Quantifying the heritability of testicular germ cell tumour using both population-based and genomic approaches." *Sci Rep* 5: 13889.

Wang, Z., K. A. McGlynn, E. Rajpert-De Meyts, D. T. Bishop, C. C. Chung, M. D. Dalgaard, M. H. Greene, R. Gupta, T. Grotmol, T. B. Haugen, R. Karlsson, K. Litchfield, N. Mitra, K. Nielsen, L. C. Pyle, S. M. Schwartz, V. Thorsson, S. Vardhanabhuti, F. Wiklund, C. Turnbull, S. J. Chanock, P. A. Kanetsky and K. L. Nathanson (2017). "Meta-analysis of five genome-wide association studies identifies multiple new loci associated with testicular germ cell tumor." *Nat Genet* 49(7): 1141-1147.

Please find our response to each of the second sets of comments from Reviewer #1.

1. *“There is still a data discrepancy. Log2 variant burden is shown in panels 2a and 2c for the same sets of samples yes, moving from all genes to cancer hallmarks actually increased the number of variants (~1.8 to ~1.88).”*

This comment represents a misunderstanding of the data presentation, **the numbers do not reflect simple variant numbers but variant burden**. We define variant burden as:

“variant burden of selected genes = total number of gHFI variants in the regions of selected genes divided by total length of the selected genes (MB).”

When we change the pool of selected genes for which we calculate the variant burden (i.e. all genes, versus cancer hallmark genes, versus hallmark minus known cancer predisposition genes), both numerator and denominator values change. Figure 2a and 2c, show that the density of gHFI variants in the area of cancer hallmark genes is higher than the density of gHFI variants in the area of all genes, not the number of variants itself.

2. A. *“There is also an unexplained data discrepancy in PCAWG, where the number of log2 number of variants is reported to be ~1.75 and then even lower to ~1.72 in the UKBB dataset. Why should there be different number of deleterious variants in these populations.”*

The main reason of this discrepancy is that sequencing protocols and variant calling pipelines are quite different between TCGA, PCAWG, and UKBB. The average exome-sequencing depth of TCGA normal sample is approximately 100X. The PCAWG data in our analysis are derived from whole-genome sequencing with the mean read depth equal to 39X in normal samples. The depth of UKBB exome-sequencing data exceeds 20X at 94.6% of exome capture sites on average. TCGA and PCAWG pancancer consortium used the same pipeline for germline variants calling (Huang, *et al.* Cell 2018), but UKBB used a different Functionally Equivalent (FE) pipeline (<https://www.ncbi.nlm.nih.gov/pubmed/30279509>). To draw attention to these important differences we have added the following section to the Methods section:

“The sequencing protocols and variant calling pipelines are different between TCGA, PCAWG, and UKBB. The average exome-sequencing depth of TCGA normal sample is approximately 100X. The PCAWG data in our analysis are derived from whole-genome sequencing with the mean read depth equal to 39X in normal samples. The depth of UKBB exome-sequencing data exceeds 20X at 94.6% of exome capture sites on average. TCGA and PCAWG pancancer consortium used the same pipeline for germline variants calling (Huang, et al. Cell 2018), but UKBB used a different Functionally Equivalent (FE) pipeline.”

2. B. *“Yet more oddness is that the number of mutations seems to have gone up with removal of the cancer risk genes.”*

This is the same misunderstanding that motivated comment one. We are not showing mutation counts. The Y-axis denotes variant burden (number of variants per MB), that is number of variants adjusted by gene length.

We generated the figure below based on TCGA data to illustrate this "oddness". If we compare the number of variants directly, the entire cancer hallmark gene set indeed has more variants than the subset of hallmark genes without the known cancer risk genes as the reviewer, and common sense, would predict (this is illustrated by the shift on Figure A). When we correlate the variant burden, across the same two gene sets as on Figure B, this is no longer seen. Most of TCGA samples fall within the red oval and have a higher variant burden in cancer hallmark without cancer risk genes compare to all cancer hallmark genes. In Figure C, we further compare the variant burden of the "cancer hallmark genes without the cancer risk genes" with the "50 cancer risk genes" in all TCGA cases. On average, the cancer hallmark gene set without cancer risk genes has a higher variant burden than known cancer risk genes, but a small group of individuals have a relatively higher variant burden of cancer risk genes. (In Figure A/B/C, each dot represents a TCGA individual, "***" indicates Wilcoxon test p value less than 0.001).

We did not include this technical background figure with the manuscript but if it helps clarify the relationship between these various definitions we are happy to include it as a supplementary figure.

4. "I don't think Figure 4 adds much to the story. I have read it and think it means nothing."

We respectfully would like to disagree with this comment. Figure 4 is very germane to our intended message that it is the combined effect of germline and somatic mutations in "cancer pathways" that lead to malignant transformation even in cancers that carry no single high penetrance germline mutations or display a strong family history. The figure illustrates this point rather clearly and intuitively by showing the contribution of germline variants and somatic mutations to alterations in 21 cancer hallmark pathways in the TCGA cancer samples.

5. "The supplements should be done without known driver genes. Ovarian should go away."

We re-calculated the association of gHFI variant burden across the age groups without cancer risk genes for individual cancer type, as shown below. A strong negative correlation still exists between average gHFI variant burden and average age of diagnosis in ovarian cancer. Because this has not changed the results, we did not include it in the paper, but happy to add as a supplementary or figure or replace the current main figure.

6. "Suppl 4 all results should be FDR corrected."

We added Holm's procedure corrected p value (p-adj) to the Suppl Figure 4, that is shown below.